# Target-Centered Drug Repurposing Predictions of Human Angiotensin-Converting Enzyme 2 (ACE2) and Transmembrane Protease Serine Subtype 2 (TMPRSS2) Interacting Approved Drugs for Coronavirus Disease 2019 (COVID-19) Treatment through a Drug-Target Interaction Deep Learning Model

**DOI:** 10.3390/v12111325

**Published:** 2020-11-18

**Authors:** Yoonjung Choi, Bonggun Shin, Keunsoo Kang, Sungsoo Park, Bo Ram Beck

**Affiliations:** 1Deargen, Inc., Daejeon 34051, Korea; yoonjungc@deargen.me (Y.C.); bonggun.shin@deargen.me (B.S.); sspark@deargen.me (S.P.); 2Department of Microbiology, College of Natural Sciences, Dankook University, Cheonan 31116, Korea; kangk1204@dankook.ac.kr

**Keywords:** COVID-19, SARS-CoV-2, coronavirus, deep learning, drug repurposing, ACE2, TMPRSS2

## Abstract

Previously, our group predicted commercially available Food and Drug Administration (FDA) approved drugs that can inhibit each step of the replication of severe acute respiratory syndrome coronavirus 2 (SARS-CoV-2) using a deep learning-based drug-target interaction model called Molecule Transformer-Drug Target Interaction (MT-DTI). Unfortunately, additional clinically significant treatment options since the approval of remdesivir are scarce. To overcome the current coronavirus disease 2019 (COVID-19) more efficiently, a treatment strategy that controls not only SARS-CoV-2 replication but also the host entry step should be considered. In this study, we used MT-DTI to predict FDA approved drugs that may have strong affinities for the angiotensin-converting enzyme 2 (ACE2) receptor and the transmembrane protease serine 2 (TMPRSS2) which are essential for viral entry to the host cell. Of the 460 drugs with *K_d_* of less than 100 nM for the ACE2 receptor, 17 drugs overlapped with drugs that inhibit the interaction of ACE2 and SARS-CoV-2 spike reported in the NCATS OpenData portal. Among them, enalaprilat, an ACE inhibitor, showed a *K_d_* value of 1.5 nM against the ACE2. Furthermore, three of the top 30 drugs with strong affinity prediction for the TMPRSS2 are anti-hepatitis C virus (HCV) drugs, including ombitasvir, daclatasvir, and paritaprevir. Notably, of the top 30 drugs, AT1R blocker eprosartan and neuropsychiatric drug lisuride showed similar gene expression profiles to potential TMPRSS2 inhibitors. Collectively, we suggest that drugs predicted to have strong inhibitory potencies to ACE2 and TMPRSS2 through the DTI model should be considered as potential drug repurposing candidates for COVID-19.

## 1. Introduction

Coronavirus disease 2019 (COVID-19) pandemic caused by the severe acute respiratory syndrome coronavirus 2 (SARS-CoV-2) [1] has become a serious threat to public health management. At least 55 million cases have been reported in more than 210 countries and territories for about eight months since December 2019, and the COVID-19 outbreak is currently in progress (Worldometer.info, 17 November 2020). While it is urgent to develop vaccines and treatments to prevent the spread of viral infections, unfortunately, no effective drug or therapy has been found for COVID-19 yet despite extensive efforts by researchers around the world except remdesivir (GS-5734). Remdesivir, developed as a treatment for Ebola virus disease, has been approved for emergency use in patients with severe COVID-19 by U.S. Food and Drug Administration (FDA) [2]. More recently, only remdesivir, now commercially named as Veklury, has been approved as the first treatment for COVID-19 by FDA (FDA press release, www.fda.gov, 22 October 2020). Multinational randomized clinical trials show that remdesivir does not make a significant difference in mortality, but is known to shorten the recovery period for patients [3]. However, more effective medications are still necessary for patients suffering from symptoms of SARS-CoV-2 infection.

Our group previously predicted several antiviral (atazanavir, remdesivir, lopinavir/ritonvir) and non-antiviral drugs (rapamycin, tiotropium) that might inhibit SARS-CoV-2 using a new deep drug–target interaction (DTI) prediction model called Molecule Transformer (MT)-DTI [4]. In the previous analysis, these drugs showed strong binding affinity values for viral proteins of SARS-CoV-2. Indeed, several antiviral drugs, such as atazanavir and remdesivir, predicted by MT-DTI, showed potent in vitro and in vivo inhibitory effects on SARS-CoV-2 infection [5,6]. Recently, we further examined that top-ranked non-antiviral drug, tiotropium bromide, has an inhibitory effect on some of the SARS-CoV-2-induced genes through transcriptome-based analysis [7]. While analyzing tiotropium bromide, it was suggested that there is a possibility of more therapeutic targets and options for COVID-19 that may be predicted by deep learning technology to fight against the on-going pandemic. To carry out the task, angiotensin-converting enzyme 2 (ACE2) and transmembrane protease serine subtype 2 (TMPRSS2) that the most studied host targets at the moment, were selected.

SARS-CoV-2 uses the ACE2 receptor and TMPRSS2 for infection into host cells [8,9]. ACE2, which serves as a binding receptor of the spike protein of SARS-CoV-2, involves in blood pressure regulation and cardiovascular regulation through enzymatic hydrolysis of angiotensin (Ang) II to Ang (1–7). Ang II cleavage by ACE2 counteracts the function of ACE that promotes Ang II formation [10,11]. Interestingly, since ACE2 is already known as a binding receptor for SARS-CoV [12], it supports that SARS-CoV-2, which maintains a very close evolutionary relationship with SARS-CoV, also utilizes ACE2 for host cell entry. Another proposed target protein, TMPRSS2, is a cell membrane-bound serine protease [13] which function is yet unclear since its identification, but the involvement of TMPRSS2 during SARS-CoV-2 entry is suggested as the host protease for the viral spike-ACE2 complex proteolytic activation [8,14].

Due to the role of ACE2 and TMPRSS2, which are essential for virus entry, they have been proposed as host targets to block SARS-CoV-2 entry [14]. Therefore, in this study, we predicted drug candidates that can control these two potential targets by using a deep-learning approach.

## 2. Materials and Methods

### 2.1. Drug Screening Database Design

Approved small molecule drugs were collected from DrugBank database [15] and listed for the drug screening database of the present study. Then, all stereoisomers of approved drugs were collected from ZINC database [16] and merged to the drug screening database by using simplified molecular-input line-entry system (SMILES) strings as queries. SMILES strings identical in both DrugBank and ZINC database were treated as approved isomers, and drugs with no SMILES match were treated as racemates (e.g., thalidomide, lenalidomide, pomalidomide, etc.). A total of 1400 drugs were selected for the screening process.

### 2.2. Affinity Prediction of Each Drug to ACE2 and TMPRSS2

A drug-target interaction deep learning model previously described as MT-DTI was used for the affinity prediction [17]. The model is pretrained with approximately 100,000,000 SMILES of chemicals in PubChem database [18] to produce a better chemical representation that considers global contexts of a given SMILES sequence through a self-attention mechanism. This self-attention mechanism proved its effectiveness in many natural language process tasks compared to local context-based models. With the pretrained MT chemical representation and a convolutional neural network (CNN) protein amino sequence representation, the MT-DTI model was trained with a combined and curated chemical-protein pairs from the Drug Target Common database [19] and BindingDB database [20] to predict affinity in *K_d_* value of given chemical-protein pairs. The performance evaluation (Concordance index, 0.882 and 0.887 in KIBA and DAVIS datasets; mean squared error, 0.152 and 0.245 in KIBA and DAVIS dataset) and comparison to other models of MT-DTI is reported in Shin et al. (2019). Finally, the screening database was processed to the model with amino acid sequences of human ACE2 (UniProt entry Q9BYF1, NCBI Reference Sequence: NP_001358344.1) and TMPRSS2 (UniProt entry O15393, NCBI Reference Sequence: NP_001128571.1) as targets.

After the prediction, results were screened for drugs with affinity *K_d_* < 100 nM to each target protein. Drugs approved as racemic mixture were excluded as they cannot be expressed in one SMILES string and may have differential ratio of isomers and biological outcomes per each isomer in the human. DTI prediction results without any filters are provided as Appendix A.

### 2.3. Cross-Prediction of MT-DTI Results through AutoDock Vina Docking Tool

AutoDock Vina (version 1.1.2), which is a molecular docking and virtual screening application [21], was used to predict binding affinities (kcal/mol) between target proteins (ACE2 [PDB ID: 1R42] or TMPRSS2) and given chemical compounds. SMILES of given chemical compounds were converted to the PDBQT format using Open Babel (version 2.3.2) [22] with the following options: --gen3d and -p 7.4. A homology model of TMPRSS2 was built using the SWISS-MODEL server (https://swissmodel.expasy.org/) due to the unavailability of the 3D structure in the protein data bank (PDB) [23]. AutoDockTools was used to convert coordinates of the given target proteins into the form needed for docking calculations by adding charges and hydrogen atoms, and merging nonpolar hydrogens (version 1.5.6) [24]. Then, binding affinities between the target proteins and given chemical compounds were calculated using AutoDock Vina. The following parameters were set for the search space: —center_x 90.654 —center_y 66.551 —center_z 39.925 —size_x 38 —size_y 92 —size_z 80 for ACE2 and —center_x 10.250 —center_y -1.306 —center_z 13.472 —size_x 28 —size_y 28 —size_z 28 for TMPRSS2. The exhaustiveness parameter was set to 10. 

### 2.4. Connectivity Map Analysis

The molecular signatures of protease inhibitors proposed as TMPRSS2 inhibitors, including bromhexine, probucol, were used to compare the gene expression signatures of the top 30 drugs in MT-DTI results for TMPRSS2 using the Touchstone database of the CLUE platform (https://clue.io) [25]. Each compound was characterized according to the connectivity score ranging from −100 to 100. Positive scores indicate the similarity between the signatures of the two compounds, and negative scores indicate the difference between the two signatures. The score corresponds to the degree of similarity or dissimilarity.

## 3. Results

To identify potent FDA-approved drugs capable of inhibiting the entry of SARS-CoV-2, we performed an in silico screening method using the MT-DTI deep learning-based model [4]. The MT-DTI model can predict the binding affinities of *K_d_* values from on chemical sequences (SMILES) and amino acid sequences (FASTA) of a target protein. The sequence-based drug-target affinity prediction approach predicted drugs with a strong binding affinity (*K_d_* < 100 nM) against ACE2 and TMPRSS2 from 1400 FDA-approved drugs (ACE2, Table 1 and Appendix A; TMPRSS2, Table 1 and Appendix A). The NCATS OpenData portal, which was developed to provide insight into drug repurposing for the treatment of COVID-19 (https://opendata.ncats.nih.gov/covid19) [26], provides drug results that affect protein–protein interaction between ACE2 and RBD (Receptor-binding domain) of the SARS-CoV-2 S protein through AlphaLISA and TruHit (counterscreen) ACE2-Spike protein–protein interaction proximity assay. We first compared the results obtained from the MT-DTI model with the drugs provided by the OpenData portal. Of the 460 drugs that may bind ACE2 with strong affinity, 17 drugs showed the interference of the interaction of ACE2 with SARS-CoV-2 spike protein (Table 2). Although AC_50_ values of the proximity assay are not affinity values, submicromolar to one-digit micromolar AC_50_ results supports the ACE2 affinity prediction of 17 selected drugs. Interestingly, enalaprilat, an ACE inhibitor, was identified as a chemical compound that has a strong affinity for ACE2 and is likely to inhibit the entry of SARS-CoV-2. According to previous results, the angiotensin-converting enzyme (ACE) and its homologous ACE2 both belong to the ACE family of dipeptidyl carboxypeptidase but provide the opposite effect for the renin-angiotensin system. ACE produces Ang II, which induces enhanced inflammation, elevated blood pressure, and increased coagulation. However, ACE2 is responsible for the breakdown and inactivation of Ang II [10,11]. Based on our predictions, enalaprilat suggests that it is a promising drug candidate that can reduce inflammation and blood pressure by inhibiting ACE activity and simultaneously inhibit the entry of SARS-Cov-2 through interaction with ACE2. Interestingly, venetoclax (*K_d_* 6.12 nM), posaconazole (*K_d_* 17.11 nM), daclatasvir (*K_d_* 6.65 nM), and ombitasvir (*K_d_* 5.91 nM) also predicted to have strong affinity for TMPRSS2, as shown in Appendix A. In particular, daclatasvir and ombitasvir are hepatitis C virus (HCV) inhibitors, suggesting that these compounds may exhibit antiviral effects on HCV and SARS-CoV-2.

Another strategy to block the entry of SARS-CoV-2 by targeting ACE2 is to increase the human recombinant soluble ACE2 (hrsACE2), thereby inhibiting the membrane association of host cells with SARS-CoV-2 [27]. Indeed, hrsACE2 has already undergone phase 1 and phase 2 clinical trials to treat acute respiratory distress syndrome and is now considered a candidate for treatment for COVID-19. In the cell, the soluble form of ACE2 is endogenously generated through ACE2 ectodomain shedding by the disintegrin metalloproteinase 17 (ADAM-17) [28]. On the other hand, calmodulin inhibits ectodomain shedding by interacting with ACE2 [29]. Therefore, inhibition of calmodulin-ACE2 interaction through the calmodulin antagonists may be an effective strategy to reduce the infectivity of the virus. We predicted that the calmodulin antagonist, the oxide of the amitriptyline, binds to ACE2 with a *K_d_* of 40.38 nM (Appendix A). However, it requires experimental verification that the drug’s interaction with ACE2 further promotes ACE2 shedding.

To compare MT-DTI affinity prediction results to widely used docking study tool AutoDock Vina, Gibbs free energy (∆G) values of each drug-target pair were predicted. For the ACE2 cross-prediction result, ∆G −8.0 kcal/mol is given as threshold, and 27 drugs show correlative prediction results out of 460 DTI predicted results (Table 3, left). The TMPRSS2 cross-prediction result was screened with ∆G −7.0 kcal/mol threshold as the numbers of predicted drugs are less than 100, and 11 out of 75 drugs showed correlation to the MT-DTI prediction (Table 3, right). However, TMPRSS2 docking results through a homology modeled TMPRSS2 may not be precise as ACE2 due to the absence of an actual 3D structure, therefore, it should be considered again when crystallography results are available. All ∆G values predicted through AutoDock Vina are annotated in Appendix A.

We have reported in the previous MT-DTI studies that tiotropium bromide has a strong binding affinity to the SARS-CoV-2 viral proteins [4]. A recent RNA-seq analysis of normal bronchial epithelium (NHBE) cells infected with SARS-CoV-2 has shown that the treatment of tiotropium can have an advantageous prescription effect [7]. Interestingly, tiotropium bromide was also predicted to strongly bind with ACE2 (*K_d_* 0.92 nM) in the present study, suggesting that the drug is a promising candidate for COVID-19.

We further investigated 75 drugs predicted to have an affinity of less than 100 nM for TMPRSS2 that is a serine protease for S protein priming of SARS-CoV-2. Clinically proven protease inhibitors, including bromhexine, aprotinin, camostat, and nafamostat, have been suggested as potential treatment options for COVID-19 [30,31]. However, the MT-DTI results for TMPRSS2 showed affinity of more than 100 nM for these protease inhibitors. Intriguingly, we found that five of the top 30 drugs that strongly bind to TMPRSS2 are antiviral drugs with a *K_d_* value of less than 20 nM (Appendix A). Three out of five (ombitasvir, daclatasvir, and paritaprevir) are anti-hepatitis C virus (HCV) drugs, and recently, there are reports that antiviral agents used to treat HCV could be another option for SARS-CoV-2 treatment [32]. Although HCV and coronavirus are not closely related viruses, it can be suggested that those drugs may act on life cycles of both viruses in a mode of action perspective as they are single-stranded RNA viruses. These three anti-HCV drugs predicted to have *K_d_* values of less than 10 nM for TMPRSS2, implicating a strong inhibitory potency (Table 1).

To screen for more effective drugs that can inhibit TMPRSS2, we used the connectivity map (CMap) database [33] that allows gene-expression signature matching between the drugs proposed as TMPRSS2 inhibitors and the top 30 drugs of the MT-DTI result. CMap is a database that provides transcriptome profiling information caused through genetic and chemical perturbations [33]. We compared gene expression changes by the previously proposed TMPRSS2 inhibitors (Bromhexine [34] and Probucol [35]) and the top 30 drugs of MT-DTI through CMap Linked User Environment (CLUE, https://clue.io) (Table 4). Scores above +90 indicate a high positive correlation between the gene-expression signature of a given perturbagen and the gene-expression signature by the query. Among the top 30 drugs, the gene expression by eprosartan and lisuride was strongly correlated with the gene expression signatures of bromhexine and probucol, respectively. Interestingly, eprosartan is a type I angiotensin II receptor (AT1R) blocker, preventing Ang II from binding to the AT1R receptor in the renin-angiotensin system [36]. As the binding of Ang II and AT1R causes fibrosis and damage of lung tissues, AT1R blocker, eprosartan, is considered a suitable therapeutic target for COVID-19 [35,36]. In addition to eprosartan, neuropsychiatric agents, lisuride, have recently been predicted as a candidate drug for treating COVID-19 through coexpression-based drug enrichment analysis of COVID-19 induced genes [37].

## 4. Discussion

An AI-integrated drug repurposing approach, such as MT-DTI, is a promising solution for the development of potential drugs that can overcome life-threatening diseases caused by COVID-19. For example, atazanavir, predicted by MT-DTI in the previous study [4], is registered in the REVOLUTIOn trial (NCT04468087) for the treatment of COVID-19. As a case study for ACE2 affinity prediction, a recent report on a clinical trial RECOVERY (NCT04381936) suggested dexamethasone, an anti-inflammatory corticosteroid, as a possible treatment option for COVID-19 [38,39]. Interestingly, dexamethasone was predicted to have an affinity of *K_d_* 9.50 nM to ACE2 (Appendix A). In addition, approved drugs sharing a similar polycyclic core structure of steroids such as obeticholic acid (*K_d_* 0.98 nM), mestranol (*K_d_* 1.02 nM), norethynodrel (*K_d_* 1.51 nM), clobetasol (*K_d_* 1.80 nM), norethisterone (*K_d_* 1.92 nM), fluoxymesterone (*K_d_* 2.29 nM), cholic acid (*K_d_* 2.86 nM), and more were listed on top in the ACE2 affinity prediction. However, original indications and side effects must be considered since steroids are involved in hormonal regulations.

Furthermore, the MT-DTI prediction resulted that ACE inhibitors, including enalaprilat, zofenopril, lisinopril, benazepril, trandolapril, cilazapril, perindopril, ramipril, fosinopril, moexipril, spirapril, have a strong binding affinity of less than *K_d_* 70 nM for ACE2 (Appendix A). Although there is concern that inhibition of ACE2 activity may promote an excessive inflammatory response resulting from an increase in Ang II [40], ACE inhibitors can be considered as a treatment option for COVID-19 in two respects. First of all, inhibition of ACE function that elicits an immune response via Ang II may alleviate the side effects of loss of ACE2 function by ACE inhibitors. Another is that ACE inhibitors do not inhibit the activity of ACE2, but rather interfere with the interaction between ACE2 and the viral S protein. Indeed, ACE2 is not only known to be insensitive to ACE inhibitors [41], but the NCATS OpenData portal provides experimental results that the enalapril maleate disrupts the protein–protein interaction between ACE2 and S protein with an AC50 of 7.5 µM. Therefore, our results suggest that the ACE inhibitor is a promising therapeutic option.

We further identified three anti-HCV drugs, including ombitasvir, daclatasvir, and paritaprevir, in the top 30 predictive candidates for TMPRSS2. Another group recently reported that anti-HCV reagents were potential drug repurposing candidates for COVID-19 [42], supporting the results. Above all, we were able to present more effective drug candidates by merging AI-based prediction results and CMap-based drug repurposing approach in this study. Eprosartan and lisuride, drugs predicted to have overall strong binding affinity values for TMPRSS2, demonstrated that they could effectively control TMPRSS2 by showing gene expression signatures similar to the previously proposed TMPRSS2 inhibitors (bromhexine, a mucolytic drug for respiratory diseases; probucol, an anti-hyperlipidemic drug). Bromhexine and probucol are known to block TMPRSS2, but both drugs are currently used as treatments for different indications and have different mode of actions other than TMPRSS2 inhibition. Consistent with this, the connectivity score between the two previously proposed TMPRSS2 inhibitors is not high. Therefore, the drugs predicted by MT-DTI do not seem to show a strong correlation for both drugs simultaneously. Interestingly, the dasatinib, which was predicted to have the strongest affinity for TMPRSS2, had little correlation with bromhexine and probucol implicating that chemical and transcriptional influence of dasatinib may be different compared to both formerly proposed TMPRSS2 inhibitors. It suggests that CMap analysis is a useful approach for filtering out possible false-positive candidates that have opposite effect or little biological effect from DTI prediction results. However, at least, any positive connectivity score, better with higher positive connectivity score, should be considered and may guide researchers to pick drugs with priority for drug repurposing.

Finally, to go deeper to target-centered drug repurposing, drugs appearing both in ACE2 and TMPRSS2 such as aclidinium, buprenorphine, emtricitabine, lurasidone, and tiotropium may be investigated as dual targeting drugs in polypharmacological perspective [43,44]. Therefore, we hope that our results can contribute to the therapeutic management of COVID-19 through evaluation efficacy and safety in vitro and in vivo studies.

Taken together, the drugs resulted through the MT-DTI affinity prediction for ACE2 and TMPRSS2 are suggested promising drug candidates for an effective drug repurposing strategy to treat COVID-19 in a target-wise perspective.

## Figures and Tables

**Table 1 viruses-12-01325-t001:** Top 20 predicted drugs identified by MT-DTI against ACE2 (NCBI Reference Sequence: NP_001358344.1) and TMPRSS2 (NCBI Reference Sequence: NP_001128571.1).

ACE2		TMPRSS2	
Small Molecules	Predicted *K_d_* in nM	Small Molecules	Predicted *K_d_* in nM
Pentostatin	0.02	Dasatinib	0.37
Liothyronine	0.43	Pentostatin	0.47
Emtricitabine	0.45	Tazemetostat	2.88
Tiotropium	0.92	Tiotropium	4.03
Obeticholic acid	0.98	Eluxadoline	5.08
Mestranol	1.02	Pimecrolimus	5.79
Gemcitabine	1.10	Tacrolimus	5.82
Levorphanol	1.30	Ombitasvir	5.91
Levallorphan	1.37	Brexpiprazole	6.06
Methscopolamine	1.40	Venetoclax	6.12
Enalaprilat	1.46	Daclatasvir	6.66
Bremelanotide	1.49	Tolvaptan	7.33
Norethynodrel	1.51	Aclidinium	8.46
Epicriptine	1.56	Paritaprevir	8.75
Rolitetracycline	1.57	Eprosartan	9.19
Diacetyl benzoyl lathyrol	1.77	Cobimetinib	10.59
Pizotifen	1.80	Entrectinib	11.08
Dapagliflozin	1.80	Lisuride	11.20
Clobetasol	1.80	Erdafitinib	11.51
Lurasidone	1.86	Letermovir	12.43

**Table 2 viruses-12-01325-t002:** The list of drugs those may affect protein–protein interaction between ACE2 and SARS-CoV-2 S protein with ACE2 *K*_d_ < 100 nM and comparison to AlphaLISA and TruHit proximity assay results provided by NCATS OpenData COVID-19.

Small Molecules	Predicted ACE2 *K_d_* (nM)	MT-DTI Rank Out of 460	Alphalisa-AC50 (μM)	TruHit-AC50 (μM)
Enalaprilat	1.46	11	7.52	-
Daclatasvir	5.81	79	5.97-8.00	4.41–6.22
Mifepristone	10.3	124	5.97	2.78
Estradiol cypionate	11.79	136	9.47	5.55
Estramustine phosphate	19.34	196	6.70	8.79
Ombitasvir	20.91	208	4.50–5.97	2.78–3.50
Norgestimate	21.06	210	9.47	4.41
d-alpha-Tocopherol acetate	21.18	212	10.62	2.21
Mitoxantrone	25.85	243	0.40–0.60	0.25–0.31
Pasireotide	28.29	257	3.67	2.69
Ergotamine	33.64	281	2.67	0.44
Flupentixol	38.26	303	8.44	4.94
Venetoclax	44.77	324	8.44	3.50
Anthralin	61.92	380	2.38	6.22
Ciclopirox	88.25	439	2.67	1.56
Thiethylperazine	88.98	440	5.66–6.70	3.12
Posaconazole	94.7	449	2.01–6.70	2.78–3.50

**Table 3 viruses-12-01325-t003:** Cross-prediction results of ACE2 and TMPRSS2 interacting drugs through MT-DTI and AutoDock Vina.

ACE2			TMPRSS2		
Small Molecules	Predicted *K_d_* (nM)	AutoDock Vina ∆G (kcal/mol)	Small Molecules	Predicted *K_d_* (nM)	AutoDock Vina ∆G (kcal/mol)
Bremelanotide	1.49	−8.1	Tazemetostat	2.88	−7.1
Talazoparib	7.04	−8	Eluxadoline	5.08	−7.5
Avapritinib	14.68	−8	Entrectinib	11.08	−7.9
Dihydroergocristine	14.94	−9.2	Erdafitinib	11.51	−7.1
Tezacaftor	15.90	−8	Aprepitant	13.52	−7.5
Dutasteride	16.35	−8.8	Canagliflozin	27.12	−7
Rifapentine	17.81	−8	Naldemedine	46.74	−7.2
Acetyldigitoxin	18.93	−8.6	Adapalene	52.49	−7.5
Alatrofloxacin	24.46	−8	Droperidol	68.38	−7.7
Deslanoside	24.75	−9.1	Larotrectinib	71.78	−7.4
Dihydroergocornine	25.99	−8.4	Zanubrutinib	75.18	−7.8
Irinotecan	28.37	−8.2			
Naldemedine	29.94	−8.8			
Ciclesonide	31.43	−8.1			
Ubrogepant	33.50	−8.1			
Ergotamine	33.64	−9.1			
Lumacaftor	36.26	−8			
Venetoclax	44.77	−9.2			
Adapalene	45.11	−8			
Letermovir	50.73	−8			
Paritaprevir	53.21	−9.2			
Entrectinib	54.40	−8.5			
Glycyrrhizic acid	61.94	−8.1			
Simeprevir	67.03	−8.7			
Glecaprevir	75.81	−8.5			
Lifitegrast	94.67	−8.3			
Posaconazole	94.70	−8			

**Table 4 viruses-12-01325-t004:** Correlation results of seven drugs with similar gene-expression patterns to TMPRSS2 inhibitors (bromhexine and probucol) among top 30 predicted drugs with TMPRSS2 *K*_d_ < 100 nM. There was no gene-expression pattern available for those drugs not listed. Bold numbers indicate connectivity score higher than 90 in the −100–100 correlation scale.

Small Molecules	Predicted TMPRSS2 *K*_d_ (nM)	Bromhexine Connectivity Score	Probucol Connectivity Score
Dasatinib	0.37	17.58	15.38
Tacrolimus	5.82	10.40	70.67
Eprosartan	9.19	**91.13**	51.17
Lisuride	11.20	34.70	**92.18**
Aprepitant	13.52	54.69	12.72
Panobinostat	15.31	33.30	41.78
Bosutinib	15.95	0.44	5.95

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
