# Peer review of "Target-Centered Drug Repurposing Predictions of Human Angiotensin-Converting Enzyme 2 (ACE2) and Transmembrane Protease Serine Subtype 2 (TMPRSS2) Interacting Approved Drugs for Coronavirus Disease 2019 (COVID-19) Treatment through a Drug-Target Interaction Deep Learning Model"

_viruses, 2020, doi:10.3390/v12111325_

Round 1
Reviewer 1 Report
Authors investigated target-centered drug repurposing prediction of ACE2 and TMPRSS2 interacting approved drugs thru Molecule Transformer-Drug Target Interaction (MT-DTI) model, a in silico model of deep learning-based drug-target interaction. Here are my comments;
Some compounds, suggested as candidates for COVID-19 treatment by in slico prediction models, failed to show their inhibitory effect of SARS-CoV-2 infection in vitro. There is still a big discrepancy between real world data and in silico prediction, so verification of MT-DTI model is needed.
- I suggest authors provide the experimental data of the essential candidate compounds to show their inhibition of ACE2-S protein binding or TMPRSS2 activity by using commercial kits and pseudotyped SARS-CoV-2-like particles in ACE2 and/or TMPRSS2-expressed cells.
- If could, authors need to demonstrate EC50 of the essential candidate compounds in SARS-CoV-2-intected cells.
In the present study, MT-DTI seems to predict the binding affinity Kd value between molecules and ACE2 or TMPRSS2. Authors should demonstrate that the predicted high affinity is directly related to the effective inhibition of ACE2-S protein binding. Among top 20 predicted drugs against ACE2, only enalaprilat is suggested as possible effective molecule on ACE2-S protein PPI.
As for similar gene-expression patterns to two TMPRSS2 inhibitors, eprosartan's scores are 91.13 and 51.17, and lisuride's scores are 34.7 and 92.18. It indicates that gene-expression patterns among TMPRSS2 inhibitors may be broad-wide, so these comparisons may be not reliable. Dasatinib, the highest affinity molecules to TMPRSS2, showed the scores 17.58 and 15.38. Authors should discuss about that.
Author Response
Responses to the reviewer 1:
Thank you for your careful review and constructive suggestions to the manuscript.
Although deep learning models and virtual docking models can be categorized as in silico prediction, there are key differences and technological advances are present in deep learning methods.
First of all, we found that there are 1) no study carrying all or most of FDA approved drug those are main molecules for drug repurposing due to limited computing power for massive in silico screening, 2) no deep learning (or machine learning) based methodology, and 3) very few studies of ACE2 and/or TMPRSS2 targeted predictions. Therefore, we claim our novelty of the study as the first ACE2 and/or TMPRSS2 drug repurposing prediction using a novel deep learning model.
Additionally, we have searched almost every in silico prediction targeting ACE2 and/or TMPRSS2, there is no deep learning based method to our knowledge. 3D-structure based docking study requires a PDB files of protein crystallography which is one of major limitation of docking simulation, TMPRSS2 prediction studies rely on homology modeling which is also predicted structure. Therefore, our 1D-string end-to-end (from sequences to affinity) model has advantages over 3D-structure based prediction. Please refer to additional information at the bottom which lists some of examples of such studies.
To respond to the reviewer 1's comments on validating MT-DTI model, the chemical space pre-training through natural language processing (called masked learning, BERT) and calculation through neural network able rapid (less than 5 minutes to predict ~2,000 molecules for one target) and precise (C-index ~0.88, ~88% accuracy respectively, Shin et al., 2019 referenced in the manuscript) prediction by the model. We included model performance in the method section with the reference in this revision. In computer science perspective, the model performance and case studies were well validated by the peers and society (Shin et al., 2019 references in the manuscript). However, respectfully, we accept and aware of the importance of laboratory experimental results. Due to the limitation of our laboratory which is computer science based (dry) lab, further investigation with experimental (wet) methods is not possible at the moment. We need to find a collaborators or build a wet lab - BSL3 will be ideal - to address the reviewer 1's point 1 and 2.
Taking the reviewer 1's considerations, we added results of docking study of ACE2, TMPRSS2 and predicted drugs and compared to the MT-DTI results with our best effort to revise. Also, external experimental data provided by NCATS opendata portal of NIH (AC50 values from AlphaLISA and TruHit ACE2-S protein proximity assays) are annotated in the revised manuscript to compensate laboratory experiment results.
Revisions made: Table 2 and 3, Line 111 to 124 (2.3 Cross-prediction of MT-DTI...parameter was set to 10) in the materials and method section, Line 181 to 190 of the result section (To compare MT-DTI affinity prediction...Vina are annotated in Table S1 and S2.)
About Connectivity map (CMap) analysis and connectivity scores, CMap analysis is based on RNA expression-response of cells to drugs. As it is not a specific drug to target interaction in a molecular level, the correlation may be low due to differential mode of actions (MoA) and off-target effects of certain drugs. Thus, CMap analysis is more like cheminformatics and transcriptomic analysis. Accordingly, bromhexine (mucolytic drug for respiratory disease) and probucol (anti hyperlipidemic drug) are different types of drugs but suggested as TMPRSS2 inhibitors during then pandemic. Therefore, connectivity scores of proposed drugs of DTI prediction may not correlate to similary to bromhexine and probucol due to different MoA and chemical structures. Additionally, the main reason of co-analysis of CMap is to increase chance of successful drug repurposing by reducing false positives as negative correlation scores will end-up with opposite biological effects as connectivity score is ranged from -100 to 100. Therefore, at least, any positive connectivity score - and, of course, better with higher positive connectivity score - may guide researchers to pick drugs with priority for drug repurposing. To respond to the reviewer 1's comment on CMap results, we added statements alerting these insight in the manuscript.
Revisions made: Line 254 to 279 (Eprosartan and lisuride, drugs predicted to...with priority for drug repurposing.)
Additional information:
For your convenience, the list below provides some examples of ACE2 interacting small molecule prediction since the beginning of pandemic (preprints are excluded), and we found that all of them used docking simulation methods (eg., autodock, glide, discovery studio, etc.) which mainly based on molecular dynamic simulations which are not in the class of machine/deep learning.
Alexpandi, R., De Mesquita, J. F., Pandian, S. K., & Ravi, A. V. (2020). Quinolines-Based SARS-CoV-2 3CLpro and RdRp Inhibitors and Spike-RBD-ACE2 Inhibitor for Drug-Repurposing Against COVID-19: An in silico Analysis. Frontiers in microbiology, 11, 1796.
Prajapat, M., Shekhar, N., Sarma, P., Avti, P., Singh, S., Kaur, H., ... & Medhi, B. (2020). Virtual screening and molecular dynamics study of approved drugs as inhibitors of spike protein S1 domain and ACE2 interaction in SARS-CoV-2. Journal of Molecular Graphics and Modelling, 101, 107716.
Gyebi, G. A., Adegunloye, A. P., Ibrahim, I. M., Ogunyemi, O. M., Afolabi, S. O., & Ogunro, O. B. (2020). Prevention of SARS-CoV-2 cell entry: insight from in silico interaction of drug-like alkaloids with spike glycoprotein, human ACE2, and TMPRSS2. Journal of Biomolecular Structure and Dynamics, 1-25.
Shanmugarajan, D., Prabitha, P., Kumar, B. P., & Suresh, B. (2020). Curcumin to inhibit binding of spike glycoprotein to ACE2 receptors: computational modelling, simulations, and ADMET studies to explore curcuminoids against novel SARS-CoV-2 targets. RSC Advances, 10(52), 31385-31399.
Choudhary, S., Malik, Y. S., & Tomar, S. (2020). Identification of SARS-CoV-2 cell entry inhibitors by drug repurposing using in silico structure-based virtual screening approach. Frontiers in immunology, 11, 1664.
Singh, N., Decroly, E., Khatib, A. M., & Villoutreix, B. O. (2020). Structure-based drug repositioning over the human TMPRSS2 protease domain: search for chemical probes able to repress SARS-CoV-2 Spike protein cleavages. European Journal of Pharmaceutical Sciences, 153, 105495.
Huggins, D. J. (2020). Structural analysis of experimental drugs binding to the SARS-CoV-2 target TMPRSS2. Journal of Molecular Graphics and Modelling, 100, 107710.
Summary of the revision
-
The model performance is mentioned in the material and methods section 2.2 with the appropriate reference.
-
Autodock vina analysis of DTI prediction results are added to the materials and methods section 2.3 and result section (Table 2). Also, same updates are done to supplementary table S1 and S2.
-
AlphaLISA and TruHit proximity assays of ACE2-Spike protein from NCATS portals are annotated in the result section (Table 3).
-
Statements and implication of CMap analysis related to dasatinib are noted in the discussion section.
-
Former table 1 and table 2 are merged to revised table 1, and SMILES sequences are deleted as they are provided in supplementary materials and not essential information in main article.
-
Minor revisions
- Significance figures are fixed to two decimals throughout the manuscript.
- Status of COVID-19 infection and recent issues on therapeutic developments are updated.
- Typo check and proof readings performed (eg., capital letters, abbreviations, wordings, italics for Kd, etc.).
Reviewer 2 Report
The authors used an already published and validated approach to treat severe acute respiratory syndrome coronavirus 2 (SARS-CoV- 2) using a home deep learning-based drug-target interaction model (MT-DTI). They used MT-DTI to predict FDA approved drug candidates that have strong binding affinities for the angiotensin-converting enzyme 2 (ACE2) receptor and the transmembrane protease serine 2 (TMPRSS2), two essential enzymes for viral entry to the host cell. As results, they identified 3 drugs with both potent ACE2 receptor and TMPRSS2 inhibitory profile. These results seem coherent with previous reported strategies targeting HCV or other viruses. Finally, they proposed the repositions of some potent FDA approved drugs.
The manuscript is well written and presented. Some details or comments concerning the binding modes (or the expected inhibition mode) of ligands would be interesting. Experimental binding of at least one compound would strengthen the strategy as well as some details in the virtual interaction between the 3 top compounds in both enzymes.
Author Response
Responses to the reviewer 2:
Thank you for your detailed review and comments on our study. To respond to the reviewer 2's suggestion, we have added docking study results and comparison with the MT-DTI result to pick out correlative drugs in the revised manuscript as well as in the supplementary material. Additionally paragraphs for methodology and results of co-prediction are given in each respective section.
Revisions made: Table 2 and 3, Line 111 to 124 (2.3 Cross-prediction of MT-DTI...parameter was set to 10) in the materials and method section, Line 181 to 190 of the result section (To compare MT-DTI affinity prediction...Vina are annotated in Table S1 and S2.)
Summary of the revision
-
The model performance is mentioned in the material and methods section 2.2 with the appropriate reference.
-
Autodock vina analysis of DTI prediction results are added to the materials and methods section 2.3 and result section (Table 2). Also, same updates are done to supplementary table S1 and S2.
-
AlphaLISA and TruHit proximity assays of ACE2-Spike protein from NCATS portals are annotated in the result section (Table 3).
-
Statements and implication of CMap analysis related to dasatinib are noted in the discussion section.
-
Former table 1 and table 2 are merged to revised table 1, and SMILES sequences are deleted as they are provided in supplementary materials and not essential information in main article.
-
Minor revisions
- Significance figures are fixed to two decimals throughout the manuscript.
- Status of COVID-19 infection and recent issues on therapeutic developments are updated.
- Typo check and proof readings performed (eg., capital letters, abbreviations, wordings, italics for Kd, etc.).
Reviewer 3 Report
This result will be predicted by many researchers in terms of mechanism of action.
However, in this study, drug candidates are derived by appropriate research design using machine learning.
I am convinced that it will be one of the important papers for COVID-19 research.
Author Response
Responses to the reviewer 3:
Thank you for your review and opinions on the manuscript. We also hope our study results and approach may boost COVID-19 therapeutics researches. We've gone through typo, grammar, and paragraph structure check and proof readings.
Revisions made: general editing works including wordings, typo, and proof readings.
Summary of the revision
-
The model performance is mentioned in the material and methods section 2.2 with the appropriate reference.
-
Autodock vina analysis of DTI prediction results are added to the materials and methods section 2.3 and result section (Table 2). Also, same updates are done to supplementary table S1 and S2.
-
AlphaLISA and TruHit proximity assays of ACE2-Spike protein from NCATS portals are annotated in the result section (Table 3).
-
Statements and implication of CMap analysis related to dasatinib are noted in the discussion section.
-
Former table 1 and table 2 are merged to revised table 1, and SMILES sequences are deleted as they are provided in supplementary materials and not essential information in main article.
-
Minor revisions
- Significance figures are fixed to two decimals throughout the manuscript.
- Status of COVID-19 infection and recent issues on therapeutic developments are updated.
- Typo check and proof readings performed (eg., capital letters, abbreviations, wordings, italics for Kd, etc.).
Round 2
Reviewer 1 Report
Authors' revision of Table 2 and 3 seems good idea. I'd like to suggest to execute PoC of MT-DTI thru SARS-CoV-2 experiment, if possible.